# Characterization of Potency of the P2Y13 Receptor Agonists: A Meta-Analysis

**DOI:** 10.3390/ijms22073468

**Published:** 2021-03-27

**Authors:** Chrisanne Dsouza, Svetlana V Komarova

**Affiliations:** 1Department of Experimental Surgery, McGill University, Montreal, QC H3G 1A4, Canada; chrisanne.dsouza@mail.mcgill.ca; 2Shriners Hospital for Children, Montreal, QC H4A 0A9, Canada; 3Faculty of Dentistry, McGill University, Montreal, QC H3A 1G1, Canada

**Keywords:** P2Y13, ADP, EC_50_, meta-analysis, purinergic signaling

## Abstract

P2Y13 is an ADP-stimulated G-protein coupled receptor implicated in many physiological processes, including neurotransmission, metabolism, pain, and bone homeostasis. Quantitative understanding of P2Y13 activation dynamics is important for translational studies. We systematically identified PubMed annotated studies that characterized concentration-dependence of P2Y13 responses to natural and synthetic agonists. Since the comparison of the efficacy (maximum response) is difficult for studies performed in different systems, we normalized the data and conducted a meta-analysis of EC_50_ (concentration at half-maximum response) and Hill coefficient (slope) of P2Y13-mediated responses to different agonists. For signaling events induced by heterologously expressed P2Y13, EC_50_ of ADP-like agonists was 17.2 nM (95% CI: 7.7–38.5), with Hills coefficient of 4.4 (95% CI: 3.3–5.4), while ATP-like agonists had EC_50_ of 0.45 μM (95% CI: 0.06–3.15). For functional responses of endogenously expressed P2Y13, EC_50_ of ADP-like agonists was 1.76 μM (95% CI: 0.3–10.06). The EC_50_ of ADP-like agonists was lower for the brain P2Y13 than the blood P2Y13. ADP-like agonists were also more potent for human P2Y13 compared to rodent P2Y13. Thus, P2Y13 appears to be the most ADP-sensitive receptor characterized to date. The detailed understanding of tissue- and species-related differences in the P2Y13 response to ADP will improve the selectivity and specificity of future pharmacological compounds.

## 1. Introduction

Purinergic signaling has gained importance as a therapeutic target since its introduction in the 1970s [1]. The family of purinergic receptors consists of two main groups, P1 receptors that include four subtypes and P2 receptors that comprise of seven P2X ligand-gated ion channels and eight P2Y G protein-coupled receptors (GPCRs). P1 receptors respond to adenosine, whereas P2 receptors are activated by adenine and uridine nucleotides, including ATP, ADP, and UTP, and UDP [2]. Purinergic signaling is relevant to the functioning of almost all body systems [3].

ADP-mediated signaling is relatively less studied compared to ATP-mediated signaling; however, it is well known for its importance for thrombosis and hemostasis. Among three P2Y receptors that are selective for ADP- P2Y1, P2Y12, and P2Y13 [4], P2Y1 and P2Y12 have been studied for their roles in platelet aggregation. P2Y1 was shown to play an important role in the initial stages of platelet aggregation, while activation of P2Y12 was demonstrated to play a critical role in regulating overall thrombosis [5,6]. The significance of P2Y12 is highlighted by several successful drugs targeting the receptor, namely clopidogrel, prasugrel, ticagrelor, and cangrelor, which are clinically used to treat cardiovascular diseases [7,8,9]. However, some studies suggested that, in addition to P2Y12, these drugs may also target the P2Y13 receptor [10,11]. Although literature on the P2Y13 receptor implicate it in various diseases [12], detailed characterization of this receptor in different tissues remains limited.

The P2Y13 receptor was discovered in 2001 [13]. It is mainly found in the brain, spleen, heart, bone marrow cells, liver, and pancreas [14]. It is structurally similar to P2Y12 and is primarily an ADP receptor [13,15]. Signaling through P2Y13 involves coupling to G_i_ proteins to inhibit adenylyl cyclase [16]. Multiple functional outcomes of P2Y13 activation have been described, such as cell proliferation, survival, and high-density lipoprotein endocytosis, as well as an important role in neuromuscular transmission, neuroprotection, and neuronal differentiation [10,17,18,19]. In mice, P2Y13 has been implicated in bone remodeling, where knockout of the receptor in mice resulted in reduced bone turnover [20].

To quantitatively summarize the existing literature on the P2Y13 receptor, we conducted a systematic search of PubMed-indexed studies reporting a concentration-dependence of P2Y13-mediated responses. Ideally, the receptor response to an agonist can be characterized by the efficacy (maximal response), affinity (the concentration at half-maximal response), and steepness (the Hill coefficient, the slope of the response). Experimentally, in each studied system, the concentration-dependence allows us to identify a system-specific maximum, EC_50_, and the steepness. Since efficacy and affinity often correlate, it is difficult to dissociate them based on the EC_50_. We assumed that, while the system-specific maximal responses are not directly comparable, the heterologously-expressed receptors demonstrate maximal efficacy, and therefore, EC_50_ obtained in those systems is indicative of the agonist affinity. Therefore, our primary objective was to assess the potency (EC_50_) of existing agonists to induce P2Y13-mediated responses. Our secondary objective was to examine if P2Y13 responses vary in different tissues and species. 

## 2. Results

### 2.1. Summary of Studies

Since we were interested in full-length peer-reviewed quantitative studies focused on the P2Y13 receptor, we used it as one of the inclusion criteria that the publication journal indexed in PubMed. The PubMed search performed on 14 September 2020 identified 153 articles focused on the P2Y13 receptor. Upon screening by two co-authors, we selected 50 articles that focused on the P2Y13 receptor for qualitative synthesis (Figure 1a). The P2Y13 receptor was studied in various physiological systems (Figure 1b), with most of the studies conducted in the nervous (20 articles) and digestive (9 articles) systems, then the musculoskeletal and circulatory systems (6 and 3 articles, respectively). In total, 10 studies employed heterologous expression systems of the P2Y13 receptor. Most studies focused on the P2Y13 receptor of murine (22 articles), human, or rat origin (14 studies each). The full-text screening identified 19 articles that examined the concentration-dependence of P2Y13-mediated responses. The concentration-dependences from one of the studies were in the linear range and therefore did not result in a good fit for the Hill curve, making it impossible to estimate the potency. Therefore, the meta-analysis was performed on 18 studies that mainly included the studies of the human, mouse, and rat P2Y13 receptor in immune, nervous, musculoskeletal, and circulatory physiological systems (Table 1). The signaling and functional responses to stimulation of P2Y13 using ADP, 2MeSADP, ADPβS, ATP, and 2MeSATP were examined in these studies (Table 1).

### 2.2. P2Y13-Mediated Signaling Events

We first focused on the responses of heterologously-expressed P2Y13. Immediate signaling events studied, following P2Y13 activation, were Ca^2+^/IP_3_ release, cAMP inhibition, and [^35^S]GTPγS binding. From nine articles, we analyzed 22 datasets [10,11,13,14,15,16,27,28,30], each reporting changes in one of the signaling outcomes in response to one agonist. If within the same article, the response to an agonist was reported multiple times, so we averaged these data. For each concentration-dependence, the individual experimental points were extracted (Figure 2a), then if the baseline data were available, it was subtracted from the data and the maximal point was scaled as 100% (Figure 2b). The resulting data were fit with the Hills curve using the Monte-Carlo method [31], allowing us to estimate the main effect size of EC_50_ values with the measure of its variance (SEM) (Figure 2c). When the dataset described P2Y13-mediated inhibition, such as for cAMP, it was inversed before fitting with the Hills equation, and IC_50_ ± SEM was identified. The forest plots represent the logEC_50/_IC_50_ ± SEM; however, in the text, we report the transformed values, EC_50/_IC_50_ ± SEM (Appendix A).

We analyzed the heterogeneity in the largest dataset of 12 concentration-dependencies for Ca^2+^/IP_3_ response. The funnel plot analysis (Figure 2d) demonstrated that the study level effects were symmetrically distributed around the random effects estimate, but did not form a funnel, suggesting that the precision in these studies did not represent the more reliable estimate of the effect size. Single study exclusion analysis (Figure 2e) identified three datasets that reduced heterogeneity; however, none of these studies affected the overall effect size. Cumulative study exclusion analysis (Figure 2f) demonstrated that 90% of studies needed to be excluded to obtain a homogenous dataset and that the effect size decreased (lower EC_50_) with a decrease in heterogeneity. 

Next, we compared the EC_50/_IC_50_ for P2Y13-mediated responses to ADP and ADP-like agonists for three signaling outcomes: Ca^2+^/IP_3_ release, cAMP inhibition, and [^35^S]GTPγS binding (Figure 3, Appendix A). Ca^2+^/IP_3_ release and cAMP inhibition required similar concentration of agonists with EC_50_ value of 9.2 nM (95% CI: 3.1–27.0) for Ca^2+^/IP_3_ release and IC_50_ of 12.3 nM (95% CI: 4.9–31.1) for cAMP inhibition. In contrast, EC_50_ for [^35^S]GTPγS binding was higher, at 136.88 nM (95% CI: 14.19–1320.78); however, the dataset for this parameter was relatively small. ADP and ADP-like agonists displayed distinct potencies in initiating different P2Y13-mediated signaling events. 2MeSADP had a lower EC_50_ of 1.93 nM (95% CI: 1.18–3.16) than ADP (EC_50_: 12.72, 95% CI: 3.0–54.5) in generating a Ca^2+^/IP_3_ response; however, for the cAMP inhibition, 2MeSADP IC_50_ was higher at 38.9 (95% CI: 29.1–52.0) compared to ADP, which had IC_50_ of 5.2 (95% CI: 1.1–25.8). In [^35^S]GTPγS binding, both 2MeSADP and ADP had wide confidence intervals, with the ADP and 2MeSADP subgroups being dominated by the Fumagalli et al. 2004 study [14]. Within studies that assessed both ADP and 2MeSADP-mediated responses within the same system, three studies reported higher potency for ADP compared to 2MeSADP [13,14,28], while two studies reported the reverse [15,30]. We also analyzed the Hills coefficient (steepness) of the agonist’s concentration-dependent curves in the largest signaling dataset of Ca^2+^/IP_3_ release (Figure 4). The steepness was similar for all ADP and ADP-like agonists, with an overall steepness of S = 4.4 (95% CI: 3.3–5.4). Overall, when all signaling responses to all ADP-like agonists were combined, the EC_50_ value for P2Y13-mediated signaling was 17.2 nM (95% CI: 7.69–38.48).

### 2.3. P2Y13-Mediated Functional Events

Next, we examined functional responses reported to be mediated by P2Y13 (Figure 5). We analyzed 14 datasets from 9 articles [17,18,21,22,23,24,25,26,29] that described functional events consisting of intermediary cellular signaling, such as Hex release [25], calcium channel inhibition [29], phosphorylation of ERK 1/2 [24], and cellular proliferation [22]. In all but one dataset [17], the functional responses were examined in the systems with endogenously-expressed P2Y13. The EC_50_ of ADP and ADP-like agonists-induced functional responses were many orders of magnitude higher than immediate signaling by heterologously-expressed P2Y13. ADP and ADP-like agonist-mediated intermediary signaling and protein activity exhibited EC_50_ of 2.3 μM (95%CI: 0.1–143.4) and 0.1 μM (95% CI: 0.01–1.87), respectively, while cellular proliferation had an even higher EC_50_ of 46.7 μM (95% CI: 4.1–532.5). Comparison of different agonists was difficult because often only few datasets were available for each agonist. Nevertheless, for receptor-mediated intermediary signaling and protein phosphorylation, 2MeSADP demonstrated the lowest EC_50_ (Figure 5, Appendix A), while ADP and ADPβS had relatively higher EC_50_ for each functional outcome (Figure 5, Appendix A). Overall, when all functional responses to all ADP-like agonists were combined, the EC_50_ value for P2Y13-mediated functional responses was 1.76 μM (95% CI: 0.3–10.06). 

### 2.4. ATP-Mediated P2Y13 Signaling

We examined available data for ATP and 2MeSATP-induced P2Y13-mediated responses (Figure 6). To be able to compare ATP-and ADP-mediated effects, we limited the dataset to studies that examined immediate signaling outcomes of heterologously-expressed P2Y13 [15,30]. ATP and 2MeSATP had an overall EC_50_ of 0.45 μM (95% CI: 0.06–3.15) to induce P2Y13-mediated signaling, which was much weaker than ADP and ADP-like agonists that signal via P2Y13 in the nanomolar range (Appendix A). 

### 2.5. P2Y13-Mediated Effects in Different Tissues

We next examined if the type of tissue studied affected the responses of endogenously-expressed P2Y13 (Figure 7). In blood tissue, functional events were studied in 4 datasets across 3 articles [21,25,26] and the EC_50_ value was 17.9 μM (95% CI: 0.8–426). In the brain, 5 datasets across 3 articles [17,22,24] examined functional responses and the EC_50_ value for ADP and ADP-like agonists was 0.3 μM (95%CI: 0.02–4.89). Thus, P2Y13-mediated responses are noticeably different in different tissues.

### 2.6. Species Origin of the P2Y13 Receptor

Finally, we assessed whether the potency of ADP and ADP-like agonists vary depending on the species origin of the P2Y13 receptors (Figure 8). We analyzed 35 datasets divided into two major categories, humans with 19 datasets from 9 articles [10,11,13,15,16,26,27,29,30] and rodents that included 16 datasets from 8 articles on mouse and rat P2Y13 [14,17,18,21,22,24,25,28]. In humans, signaling outcomes required nM levels of agonists with the order of potency of 2MeSADP > ADP > ADPβS, while in rodents, ADP and 2MeSADP had similar potency (Figure 8, Appendix A). Overall, ADP and ADP-like agonists displayed a lower potency to induce signaling events in human P2Y13, with EC_50_ of 7.4 nM (95% CI: 2.9–18.8), compared to rodent P2Y13, with EC_50_ of 149.8 nM (95% CI: 64.3–348.8). In humans, functional responses could not be divided into agonist types due to a limited number of studies. In rodents, the EC_50_ for P2Y13-mediated functional responses was lower for 2MeSADP compared to ADP and ADPβS (Figure 8, Appendix A). Overall, the functional responses for rodent P2Y13 displayed lower EC_50_ than for human P2Y13 (Figure 8, Appendix A).

## 3. Discussion

We systematically identified 18 studies that assessed the concentration dependence of the P2Y13 receptor and quantitatively summarized the EC_50_ of P2Y13 responses to different agonists. We found that ADP and ADP-like agonists are three orders of magnitude more potent than ATP and ATP-like agonists at the P2Y13 receptor. 2MeSADP was more potent than ADP or ADPbS for immediate signaling events in pharmacological assays, in particular for human P2Y13, but exhibited similar potency for physiological signaling events, such as cAMP inhibition. We found that much lower agonist concentrations are required to elicit immediate signaling events in heterologous expression systems, such as calcium and cAMP signaling, compared to functional outcomes in endogenous expression systems, such as protein phosphorylation and cell proliferation. Finally, our data suggest that ADP and ADP-like agonists have a lower EC_50_ at the brain P2Y13 compared to the blood P2Y13 and an order of magnitude lower EC_50_ at the human P2Y13 compared to rodent P2Y13. 

We provide the quantitative estimates for the potency of ADP, ATP, and their derivatives in stimulating P2Y13-mediated responses. In heterologous systems, which we assumed provide a maximal efficacy for P2Y13-mediated effects, our meta-analysis shows that 2MeSADP was more potent in a pharmacological assay that relies on activation of Ca^2+^/IP_3_ release, while ADP was equivalent or more potent in cAMP inhibition, which is the physiological outcome of this Gi-coupled receptor [15]. Previous studies on the P2Y13 receptor similarly concluded that, for P2Y13, the difference in potency between 2MeSADP and ADP depends on the experimental conditions and in most cases, is not as pronounced as for the P2Y12 receptor [13,15,30]. In keeping with previous studies [15,29,30], our meta-analysis demonstrates that ATP and 2MeSATP act as weak agonists of the P2Y13 receptor. The EC_50_ for P2Y13-mediated inhibition of cAMP by ADP was estimated as 5.24 nM (95% CI: 1.06–25.80), which confirms that ADP has a higher potency at the P2Y13 receptor compared to other ADP-sensitive receptors, P2Y1 and P2Y12 [4]. 

We found that the reported EC_50_ for ADP and ADP-like agonists of P2Y13 was consistently higher (by 3 orders of magnitude) in the studies that focused on functional responses of the endogenously-expressed receptor, such as cellular proliferation and protein phosphorylation, compared to signaling outcomes of P2Y13 activation in heterologous systems. There could be several potential explanations for this observation. First, it is important to note that the majority of the signaling or pharmacological studies used the heterologously-expressed P2Y13 receptor, often co-transfected with chimeric G proteins, such as G_⍺/16_, to isolate its responses from P2Y1 and P2Y12, while the endogenously expressed receptor was used in studies of functional outcomes. Since EC_50_ is a complex function of the affinity for the receptor and its efficacy, which depends on the expression level [32], the apparent EC_50_ would be expected to be lower for the exogenously-expressed receptor compared to the endogenously expressed receptor. However, the data quantifying the absolute amounts of receptors present in different systems are not available to validate this possibility. Second, the timing between signaling events measured immediately upon the addition of ligand and functional outcomes, which usually take hours or days, may lead to the differences in observed EC_50_. It is possible that a sustained agonist application is needed to observe similar EC_50_ in functional outcomes. Finally, it is possible that cooperativity with other receptors is required to induce a functional response through P2Y13, since, naturally, ADP is presented not in isolation, but rather in combination with ATP [33]. Since no study investigated both signaling and functional events, future studies are required to uncover the mechanisms underlying these differences.

Our study suggests that P2Y13 properties may depend on the tissue and species origin of the receptor. Specifically, the subgroup analysis that took into account agonist and outcome type suggested that ADP and its agonists are more potent at the brain P2Y13 than the blood P2Y13 and that ADP and ADP-like agonists are more potent at the human P2Y13 receptor than the mouse P2Y13 receptor. While these findings are potentially interesting, they require further confirmation, since subgroup analysis resulted in a relatively low number of datasets per condition. No study has directly compared P2Y13 responses in different tissues, which is of potential interest when off-target tissue pharmacological side effects are considered. Moreover, the expression levels of P2Y13 can differ in different tissues, resulting in variability in the apparent potency of its agonists. In regard to the species origin of the receptor, the original study by Zhang and co-authors [15] demonstrated that ADP and its analogs have higher potency at murine P2Y13 compared to human P2Y13, while we show that rodent receptors (mouse and rat) demonstrate higher EC_50_ compared to human P2Y13 in signaling responses. Therefore, further comparative studies for receptors of murine, rat, and human origin are needed to identify their differences, which may translate to distinct in vivo responses to related pharmacological compounds in different species. 

Several limitations of our study should be noted. The majority of the studies used the heterologously-expressed P2Y13 receptor; therefore, we were limited in studying the endogenous response of the P2Y13 receptor. Heterologous expression of a receptor may result in lower EC_50_ values if the population of the P2Y13 receptors is higher than in the native environment. Therefore, characterizing native P2Y13-specific downstream signaling events, such as cAMP activity, remains important. However, it should be noted that, in the absence of specific P2Y13 agonists, it is experimentally difficult to separate the P2Y13-mediated responses from those mediated by other ADP-responsive P2 receptors, P2Y12 and P2Y1, since, in many cells, all three receptors can be expressed simultaneously [34,35,36]. Another limitation related to data availability was the small number of datasets available for some experimental conditions, making the subgroup analysis difficult. For example, studies on blood cells only had functional responses and no signaling responses and the analysis of the time dependence of P2Y13 activation was not possible, since only two studies examined it. We also excluded one study from the meta-analysis because the concentration-dependence presented in that study was limited to the linear range, precluding us from extracting EC_50_, which was our main meta-analytic outcome. Finally, we observed a high level of heterogeneity in the meta-analysis. While some heterogeneity was explained by the type of agonists, the type of outcome, and the cell origin, the remaining heterogeneity in several cases was still high. This may be due to the inherently different experimental conditions across studies and will be alleviated when this relatively new receptor is characterized more thoroughly. 

In conclusion, our meta-analysis provides the EC_50_ estimates with confidence intervals for different outcomes of P2Y13-mediated responses. We have shown that the potency of agonists on P2Y13 differs according to the outcome, type of tissue, and the origin of species. These studies provide an important step towards quantitative characterization of different P2Y receptors. In the future, this will allow differentiating P2Y13-mediated effects from those induced by P2Y12 or P2Y1. P2Y12-targeting compounds, such as clopidogrel, ticagrelor, and cangrelor, are now in clinical practice [37,38], and it was previously suggested that these drugs may also interfere with P2Y13 [10,30]. Therefore, the detailed understanding of different receptors will help in improving the selectivity and specificity of future pharmacological compounds.

## 4. Materials and Methods 

This study implemented the preferred reporting items for systematic review and meta-analysis (PRISMA) checklist [39]. 

### 4.1. Software

EndNote X9 (Clarivate Analytics, Philadelphia, PA, USA) was used to import research articles from PubMed. Data extraction and meta-analysis were carried out using the META-LAB plug-in for MATLAB R2016b (The MathWorks Inc., Natick, MA, USA) [31]. Interpolation and normalization of data were performed on GraphPad Prism 9 (SanDiego, CA, USA). Results obtained from the above-mentioned software were exported to Microsoft Excel for further characterization and analysis. The plots generated from META-LAB and Microsoft Excel were edited using CorelDraw Graphics Suite 2020 (Ottawa, ON, Canada). 

### 4.2. Summary of Studies 

The search strategy was developed such that the resulting studies contained concentration-dependence curves of the agonists to the P2Y13 receptor (Appendix A). The search was conducted on 3 July 2019 on PubMed and was updated on 14 September 2020, as the database covers an extensive list of high-quality articles, and the results were exported to EndNote. Reviews, editorials, letters, and conference abstracts were excluded and only papers in English were included. Abstracts were screened by two independent reviewers (C.D. and S.V.K.) to identify papers focusing on the P2Y13 receptor and its function. Selected abstracts were further full text screened by the experiments conducted in the study. Only concentration-dependence studies of agonist(s) on the P2Y13 receptor were included, while others were excluded. The selected studies were assessed for their quality using a questionnaire (Appendix A).

### 4.3. Data Extraction 

The figures from selected studies were provided as input to META-LAB’s “Data Extraction” module, which facilitated the extraction of concentration-dependent experimental values and their reported variance. When provided, control or baseline experiments were also extracted. The sample size and the type of variance (standard error or standard deviation) were recorded. When the measure of variance was not reported, it was assumed to be a standard error. Studies with unreported sample size were set to 3. In cases of studies providing a range of sample size, the lowest number was selected. 

### 4.4. Study Level Outcomes 

Study level outcomes included (i) EC_50_. The concentration of agonist required to achieve a half-maximum response and (ii) steepness of the slope (S) in the largest dataset in P2Y13-mediated signaling responses (Ca^2+^/IP_3_ release). 

### 4.5. Scaling and Normalization of Data 

Before fitting, the extracted data were scaled and normalized such that it ranged from 0–100% of output measurement:$$A_{\mathrm{normalized}}=\;\frac{A-\mathrm{minimum}\;\mathrm{value}}{\mathrm{maximum}\;\mathrm{value}-\mathrm{minimum}\;\mathrm{value}}\;\%$$

When baseline measurements were provided, concentration-dependent values were subtracted from the baseline prior to normalizing with the highest value to obtain measurements pertaining to the P2Y13. When the baseline was absent, the lowest value was selected as the minimum response.

### 4.6. Fitting Dose-Dependency Curves in Hill Function 

Using META-LAB’s “Fit module”, the normalized data were fit to the Hill equation with the Monte-Carlo error propagation method and iterated to 1000 times to obtain EC_50_ and the steepness of the slope. The equation used: $$Y=\frac{\beta_{1}X^{\beta_{3}}}{\beta_{2}^{\beta_{3}}+\;X^{\beta_{3}}}$$
where $\beta_{1}$ is the maximum response, $\beta_{2}$ is the agonist’s concentration required to induce half of its maximal response, and $\beta_{3}$ is the steepness of the slope.

### 4.7. Preparing Data for Meta-Analysis 

Multiple independent measurements of the same agonist within studies were averaged for the analysis. Covariates were assigned to individual studies to conduct subgroup analysis. 

### 4.8. Meta-Analysis 

A random-effects model was used to pool study estimates, EC_50_, and steepness (S) to generate a summary outcome $\hat{\theta }$:$$\hat{\theta }=\;\frac{\sum_{i}^{N}\left(\theta_{i}\;\times \;w_{i}\right)}{\sum_{i}(w_{i})},\;$$
where N refers to the number of datasets.

The standard error computed for the summary effect:$$se\;\left(\hat{\theta }\right)=\;\frac{1}{\sqrt{\sum_{i}^{N}w_{i}}}$$

The 95% CI was computed using a z-distribution: $$CI=\;\pm z_{\left(1-\frac{\alpha }{2}\right)},\;$$
where α = 0.05.

The weights $w_{i}$ assigned to each study were calculated using an inverse variance approach:$$w_{i}=\;\frac{1}{se{\left(\theta_{i}\right)}^{2}+\;\tau^{2}}$$

The inter-study variance was predicted using the DerSimonian-Laird estimator:$$\tau^{2}=\;\frac{Q-\left(N-1\right)}{c}$$
where Q is the heterogeneity statistic, N is the number of datasets, and c is the scaling factor. 

### 4.9. Subgroup Analysis 

Subgroup analysis was performed depending on (i) the type of tissue studied and (ii) the origin of the species of the studied P2Y13 receptor.

### 4.10. Heterogeneity 

The heterogeneity among data was quantified using the H^2^ statistic:$$H^{2}=\;\frac{Q}{df}$$
where Q is the measure of total variation and df is the degrees of freedom. H^2^ refers to the magnitude of heterogeneity in the data independent of the number of studies. 

Funnel plot, single study, and cumulative study exclusion analyses were performed to assess heterogeneity and publication bias. The funnel plot was generated with the inverse standard error, 1/se ($\theta_{i}$), against the individual study effect size $\theta_{i}$. The homogeneity threshold, T_H_, represents the percentage of studies that need to be removed to achieve homogeneity. 

## Figures and Tables

**Figure 1 ijms-22-03468-f001:**
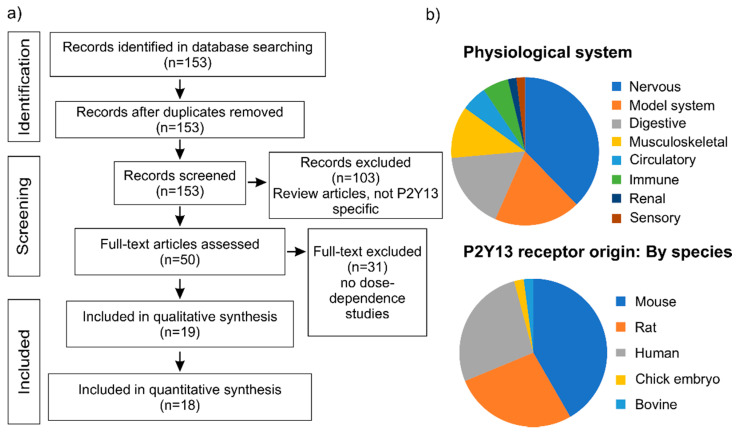
Screening and selection process. (**a**) Preferred reporting items for systematic reviews and meta-analyses (PRISMA), adapted. n indicates the number of studies (**b**) The pie chart indicates the distribution of P2Y13 studies obtained at the full-text screening level (n = 50) in different species in terms of the receptor’s origin and physiological systems. Studies performed in the heterologously-expressed P2Y13 receptor are indicated as model systems.

**Figure 2 ijms-22-03468-f002:**
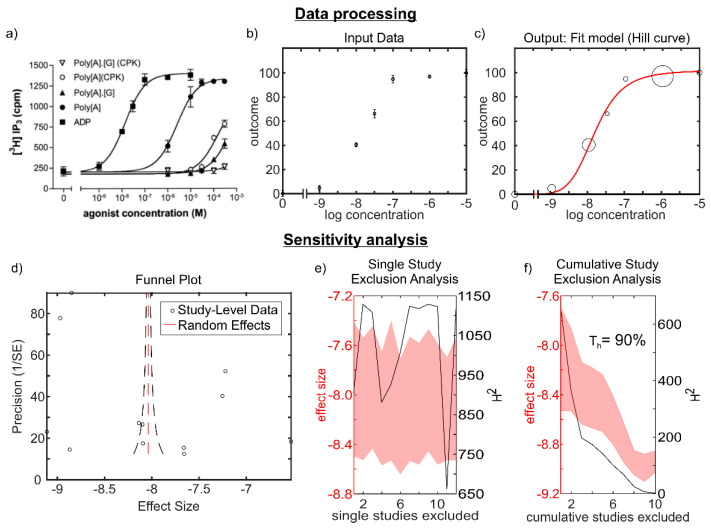
Data processing and sensitivity analysis in Ca^2+^/IP_3_ datasets. (**a**) Example of a concentration-dependence curve from a study (Marteau et al. 2003 ^#^, reproduced with permission) that was used to extract P2Y13-mediated responses. (**b**) Extracted dataset was used as an input to fit into the Hill function. (**c**) Final output of the dataset fit into the Hill curve after Monte-Carlo simulations. (**d**) Heterogeneity was analyzed in Ca^2+^/IP_3_ release dataset using a forest plot. Study-level (log10) effect sizes (black circles) were plotted against the precision (inverse standard error). The red dashed line represents the random effects estimate. (**e**) Single study and (**f**) cumulative study exclusion analysis was carried out to assess studies that contributed to heterogeneity. Red shaded region: 95% confidence interval; black line: H^2^ statistic; T_h_: homogeneity threshold ^#^ Reference. Marteau, F.; Le Poul, E.; Communi, D.; Communi, D.; Labouret, C.; Savi, P.; Boeynaems, J.M.; Gonzalez, N.S. Pharmacological characterization of the human P2Y13 receptor. *Mol. Pharmacol.*
**2003**, *64*, 104–112, doi:10.1124/mol.64.1.104 [30].

**Figure 3 ijms-22-03468-f003:**
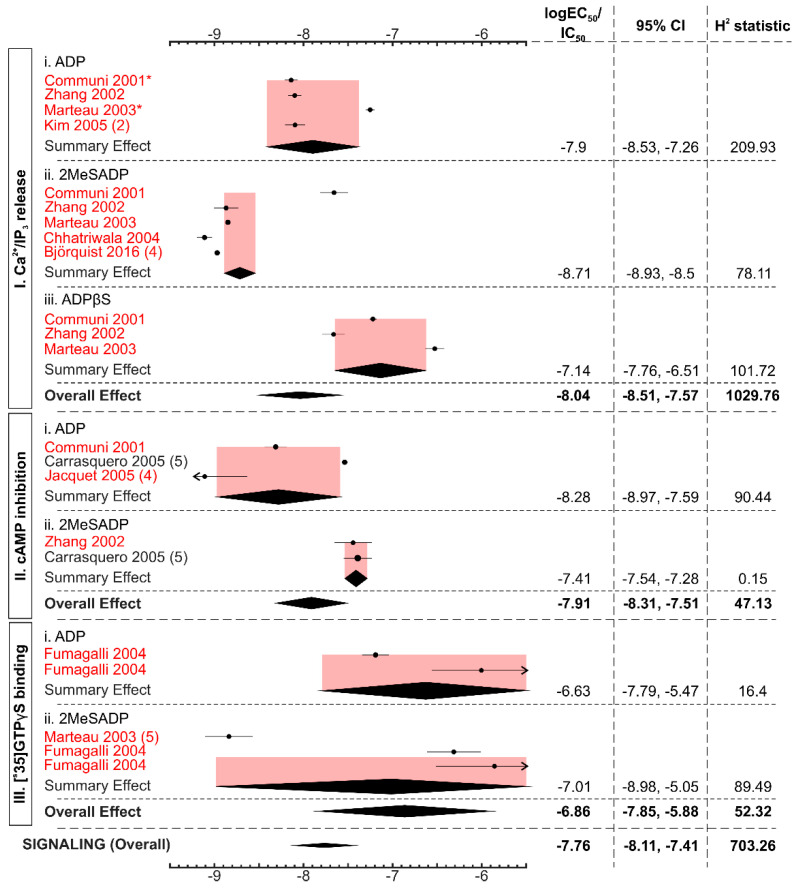
Signaling responses to ADP and ADP-like agonists in the P2Y13 receptor. The forest plot represents the log_10_ EC_50_/IC_50_ values of different ADP and ADP-like agonists to generate P2Y13-mediated signaling outcomes. The black diamond represents the overall effect size, and the pink shaded region represents the 95% confidence interval. The majority of studies were conducted in heterlogously-expressed P2Y13 (marked in red), with the exception of the endogenously expressed P2Y13 study in Carrasquero et al., 2005. All the studies were of sample size, n = 3, unless otherwise stated in parentheses. The studies marked in * had multiple recordings of the same agonist and outcome in the individual study and were averaged, with Communi et al., 2001, having 2 datasets and Marteau et al. 2003 having 3 datasets.

**Figure 4 ijms-22-03468-f004:**
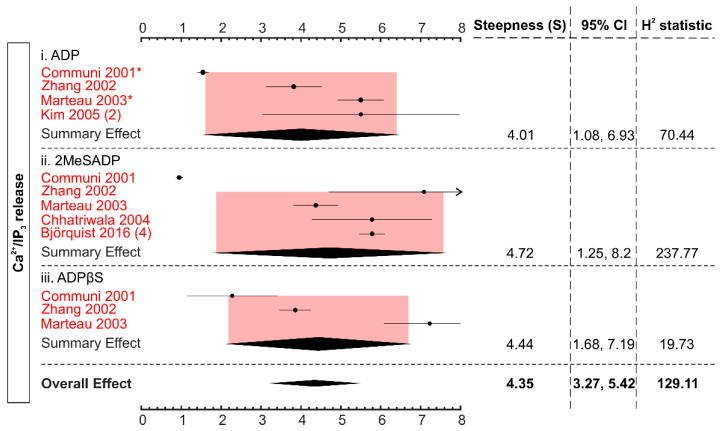
Steepness (S) in Ca^2+^/IP_3_ release dataset. The steepness of slope (S) was assessed in the largest signaling dataset, Ca^2+^/IP_3_ release. The black diamond represents the overall effect size, and the pink shaded region represents the 95% confidence interval. The study names marked in “red” were conducted in the heterologously-expressed P2Y13 receptor. All the studies were of sample size, n = 3, unless otherwise stated in parentheses. The studies marked in * had multiple datasets of the same agonist and outcome in the individual study and were averaged, with Communi et al., 2001, having 2 datasets and Marteau et al. 2003 having 3 datasets.

**Figure 5 ijms-22-03468-f005:**
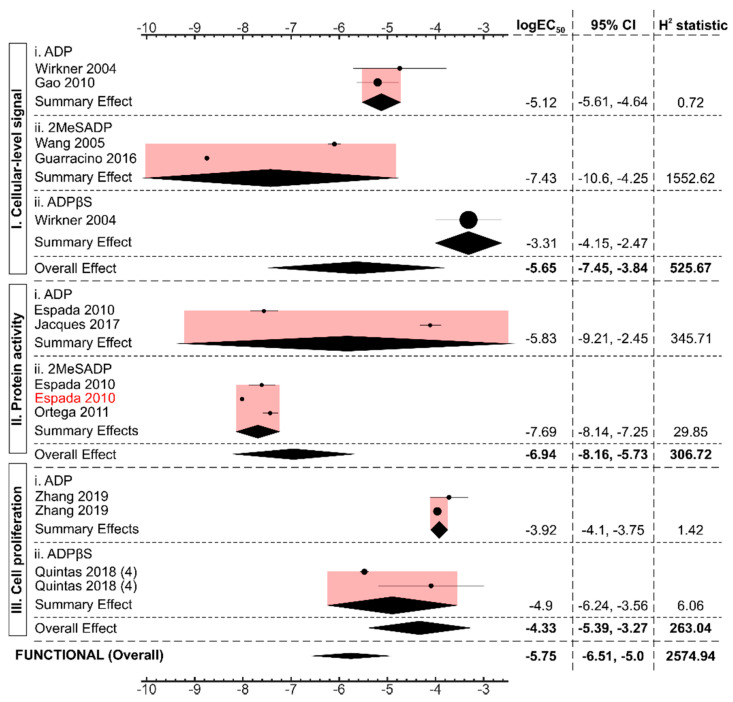
P2Y13-mediated functional responses with ADP and ADP-like agonists. Forest plot of log_10_ EC_50_ values of different ADP and ADP-like agonists to generate P2Y13-mediated functional outcomes. The black diamond represents the overall effect size, and the pink shaded region represents the 95% confidence interval. All studies were conducted in systems endogenously expressing the P2Y13 receptor with the exception of the dataset marked in “red” from the Espada et al. 2010 study, conducted in the heterologously-expressed P2Y13 receptor. All the studies were of sample size, n = 3, unless otherwise stated in parentheses.

**Figure 6 ijms-22-03468-f006:**
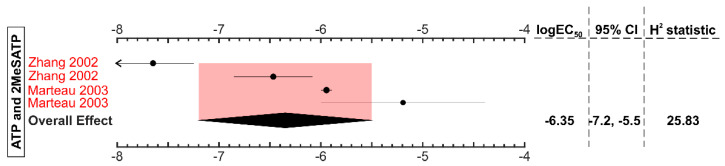
P2Y13-mediated signaling outcomes with ATP and ATP-like agonists. The forest plot contains the log_10_ EC_50_ values of ATP and 2MeSATP in generating P2Y13-mediated signaling outcomes. The black diamond represents the overall effect size, and the pink shaded region represents the 95% confidence interval. The study names marked in “red” were conducted in the heterologously-expressed P2Y13 receptor. All the studies were of sample size, n = 3, unless otherwise stated in parentheses.

**Figure 7 ijms-22-03468-f007:**
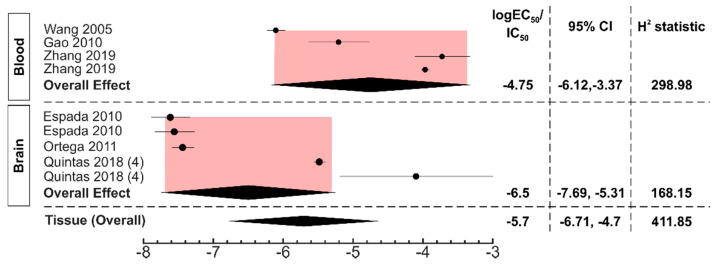
Subgroup analysis for endogenous P2Y13 responses in tissues. The forest plot represents the log_10_ EC_50_ values in the blood and brain to generate a P2Y13-mediated functional response. The black diamonds and the pink shaded region represent the overall effect size and the 95% confidence interval. All the studies were of sample size, n = 3, unless otherwise stated in parentheses.

**Figure 8 ijms-22-03468-f008:**
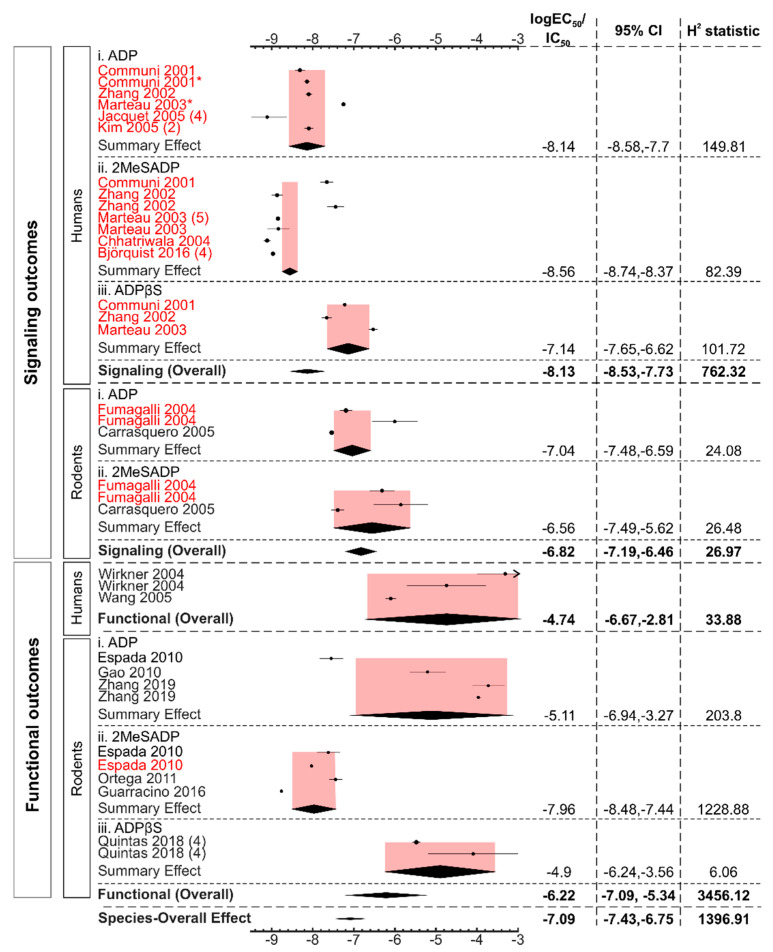
Subgroup analysis based on the species of the P2Y13 receptor. The forest plot represents the log_10_ EC_50_ values in different P2Y13 species (humans and rodents) to generate a P2Y13-mediated response. The black diamonds and the pink shaded regions represent the overall effect size and the 95% confidence interval. The study names in “red” were conducted in the heterologously-expressed P2Y13 receptor. All the studies were of sample size, n = 3, unless otherwise stated in parentheses. The studies marked in * had multiple datasets of the same agonist and outcome in the individual study and were averaged, with Communi, 2001, having 2 datasets and Marteau, 2003, having 3 datasets.

**Table 1 ijms-22-03468-t001:** Overview of selected papers for systematic review and meta-analysis. The search strategy is included in Appendix A. Each paper was assigned a quality score based on a developed questionnaire (Appendix A), where 14 points indicated a high-quality paper. The study names marked in “red” were conducted in a model system with heterologous P2Y13 expression. ADPβS: adenosine 5′-O-(2-thiodiphosphate); CREB: cAMP-responsive element binding protein; 2MeSADP: 2-methylthio-adenosine diphosphate; MEPP: miniature end-plate potentials; ERK: extracellular signal-regulated kinase; IP_3_: inositol trisphosphate; [^35^S]GTPγS: [^35^S]guanosine-5′-O-(3-thiotriphosphate); 2MeSATP: 2-methylthio-adenosine triphosphate.

Author, Year	Physiological System	Species	Agonist	Outcome(s)	P2Y13 Expression	Quality Score
Zhang et al., 2019 [21]	Immune	Mouse	ADP	Viral RNA replication, Cell viability	Endogenous	13
Quintas et al., 2018 [22]	Nervous	Rat	ADPβS	Cell proliferation	Endogenous	12
Jacques et al., 2017 [23]	Sensory	Chick embryo	ADP	p-CREB	Endogenous	13
Guarracino et al., 2016 [18]	Musculoskeletal	Mouse	2MeSADP, ADP, ATP	MEPP frequency	Endogenous	14
Björquist et al., 2016 [11]	Model system	Human	2MeSADP	Dynamic Mass Redistribution	Heterologous	13
Ortega et al., 2011 [24]	Nervous	Rat	2MeSADP	p-ERK	Endogenous	12
Gao et al., 2010 [25]	Immune	Rat	ADP	Hex release	Endogenous	14
Espada et al., 2010 [17]	Nervous Model system	Mouse	ADP 2MeSADP	Hmox1 induction	Endogenous Heterologous	14
Wang et al., 2005 [26]	Circulatory	Human	2MeSADP	ATP release, cAMP accumulation	Endogenous	14
Kim et al., 2005 [27]	Model system	Human	ADP	IP_3_ accumulation	Heterologous	12
Jacquet et al., 2005 [10]	Model system	Human	ADP	cAMP accumulation	Heterologous	14
Carrasquero et al., 2005 [28]	Nervous	Rat	2MeSADP, ADP	cAMP accumulation	Endogenous	14
Wirkner et al., 2004 [29]	Renal	Human	ADP, ADPβS, ATP	Inhibition of N-type calcium channels	Endogenous	12
Fumagalli et al., 2004 [14]	Model system	Rat	2MeSADP, ADP	[^35^S]GTPγS binding	Heterologous	14
Chhatriwala et al., 2004 [16]	Model system	Human	2MesADP	IP_3_ accumulation	Heterologous	14
Marteau et al., 2003 [30]	Model system	Human	2MeSADP, ADP, ADPβS, ATP, 2MeSATP	[^35^S]GTPγS binding, IP_3_ accumulation	Heterologous	14
Zhang et al., 2002 [15]	Model system	Human	2MeSADP, ADP, ADPβS, 2MeSATP	Intracellular calcium, cAMP accumulation	Heterologous	14
Communi et al., 2001 [13]	Model system	Human	2MeSADP, ADP, ADPβS, ATP, 2MeSATP	IP_3_ accumulation, cAMP accumulation	Heterologous	14

## Data Availability

The data presented in this study are available on request from the corresponding author.

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
