# Peer review of "Characterization of Potency of the P2Y13 Receptor Agonists: A Meta-Analysis"

_ijms, 2021, doi:10.3390/ijms22073468_

Round 1
Reviewer 1 Report
The motivation for this study is the lack of detailed characterization of the P2Y13 receptor, which is implicated in various diseases.
The objectives of this study were to assess the potency of known agonists to induce P2Y13-R mediated responses and to examine if the latter vary in different tissues and species.
Comments:
The study was well conducted using appropriate techniques. However, it is not innovative, but is rather based on analysis of previously reported data. Likewise, the analysis is performed using routine methods.
The discussion is thorough leading to invaluable conclusions regarding P2Y13-R mediated responses.
The manuscript is well written and edited.
I recommend accepting this manuscript as is.
Author Response
We thank the reviewer for their time and effort and for the positive review of our work.
Reviewer 2 Report
The present study is a very interesting work aiming to summarize P2Y13R data reported in the literature. P2Y13R is one of the metabotropic nucleotide receptors selective for ADP that shares many structural and pharmacological features with other ADP receptors especially with P2Y12R. Recently, it has been reported that P2Y12R antagonists used in cardiovascular therapy also interact with P2Y13R acting as partial agonists. These findings have attracted more attention to ADP nucleotide receptors.
Overall, the work has been well designed, but there are some concerns that need to be reconsidered.
According to the first objective proposed, authors have selected 18 papers of the literature reporting concentration response curves for agonist responses recorded from heterologous expressed and endogenous P2Y13 receptors shown in Table 1. I agree with the selection, in fact, it is well done. However, I disagree with some statements.
Authors have assigned physiological functions to P2Y13R from results obtained by heterologous expression of P2Y13 receptor in different cellular lines (HEK, CHO, astrocytoma, etc), and from my knowledge, this is not correct. I strongly recommend to eliminate data related to this statement (Diagram from figure 1, the column 2 from table 1). According to this, epigraph 2.5 and Figure 7 must be revised. This epigraph should include results obtained in native tissues (blood and brain) and studies performed with heterologous expressed P2Y13 must be removed. In addition, data reported for endogenous P2Y13R present in cerebellar granule neurons from the study of Espada et al should be added.
SPECIFIC COMMENTS
Specific comments have been included in the enclosed PDF file.

Author Response
We thank the reviewer for their time and considerable effort, and for the valuable feedback on the manuscript. As suggested by the reviewer, the following changes were implemented:
- We agree with the reviewer that it is not appropriate to use the results obtained from heterologous expression of P2Y13 to infer the physiological function. Accordingly, we revised the paper as follows:
- We re-evaluated the classification of physiological systems for Fig 1B and assigned “model system” to studies using heterologous expression models. Figure 1B is updated accordingly
- We implemented the same classification in the column 2 of Table 1 (using actual tissue for endogenously expressed receptors, and model system for heterologous expression).
- Revised tissue analysis and the corresponding Figure 7, which now only include P2Y13-mediated functional responses in blood and brain tissue
- Updated the corresponding descriptions in the abstract, results and discussion sections.
- As suggested, we now included the datasets from Espada et al that describe endogenously expressing P2Y13-mediated functional responses.
- We carefully addressed specific comments in the PDF file enclosed by the reviewer and made appropriate changes as indicated in the revised version of the manuscript with the tracked changes.
Reviewer 3 Report
In the paper of Dsouza and Komarova authors present a meta-analysis of studies related to potency of the P2Y12 receptor agonists. They search the PubMed data available and finally concentrated on 19 original papers that meet the inclusion criteria. Authors characterized efficacy of different agonists of P2Y12 receptors described in all these papers according to the functional activity and chemical structure of the agonists, tissue and species origin of the receptors. They found that generally ADP-based agonists are more potent than others, and human P2Y13 receptors are more sensitive than of other origin.
This article is the next in a series of papers emerging from Komorova's group devoted to meta-analysis of data related to basic research. The article is well written, has all the necessary attributes, research methods are described in sufficient detail, the results are new and interesting, conclusions follow from the results obtained.
I have no comments on the presented work.
The only suggestion for Figure 1b) – would be better to indicate to which group of records it is related to – to all identified (153), full text (50) or included only (19).
Author Response
We thank the reviewer for their time and effort and the positive evaluation of our manuscript. As suggested by the reviewer, we now clarify that 50 studies selected for full text analysis were analyzed in Figure 1B.